# DISTANCE-BASED LEARNING FROM ERRORS FOR CONFIDENCE CALIBRATION

**Chen Xing***
College of Computer Science,
Nankai University
Tianjin, China

**Sercan Ö. Arık**
Google Cloud AI
Sunnyvale, CA

**Zizhao Zhang**
Google Cloud AI
Sunnyvale, CA

**Tomas Pfister**
Google Cloud AI
Sunnyvale, CA

## ABSTRACT

Deep neural networks (DNNs) are poorly calibrated when trained in conventional ways. To improve confidence calibration of DNNs, we propose a novel training method, distance-based learning from errors (DBLE). DBLE bases its confidence estimation on distances in the representation space. In DBLE, we first adapt prototypical learning to train classification models. It yields a representation space where the distance between a test sample and its ground truth class center can calibrate the model's classification performance. At inference, however, these distances are not available due to the lack of ground truth labels. To circumvent this by inferring the distance for every test sample, we propose to train a confidence model jointly with the classification model. We integrate this into training by merely learning from mis-classified training samples, which we show to be highly beneficial for effective learning. On multiple datasets and DNN architectures, we demonstrate that DBLE outperforms alternative single-model confidence calibration approaches. DBLE also achieves comparable performance with computationally-expensive ensemble approaches with lower computational cost and lower number of parameters.

## 1 INTRODUCTION

Deep neural networks (DNNs) are being deployed in many important decision-making scenarios (Goodfellow et al., 2016). Making wrong decisions could be very costly in most of them (Brundage et al., 2018) – it could cost human lives in medical diagnosis and autonomous transportation, and it could cost significant business losses in loan categorization and sales forecasting. To prevent these from happening, it is strongly desired for a DNN to output confidence estimations on its decisions. In almost all of the aforementioned scenarios, detrimental consequences could be avoided by refraining from making decisions or consulting human experts, in the cases of decisions with insufficient confidence. In addition, by tracking the confidence in decisions, dataset shifts can be detected and developers can build insights towards improving the model performance.

However, confidence estimation (also referred as 'confidence calibration') is still a challenging problem for DNNs. For a 'well-calibrated' model, predictions with higher confidence should be more likely accurate. However, as studied in (Nguyen et al., 2014; Guo et al., 2017), for DNNs with conventional (also referred as 'vanilla') training to minimize the softmax cross-entropy loss, the outputs do not contain sufficient information for well-calibrated confidence estimation. Posterior probability estimates (e.g. the softmax outputs) can be interpreted as confidence estimation, but it calibrates the decision quality poorly (Gal & Ghahramani, 2016) – the confidence tend to be large even when the classification is inaccurate. Therefore, it would be desirable if the training approach is redesigned to make its decision making more transparent, thus providing more information for accurate confidence calibration.

In this paper, we propose a novel training method, Distance-based Learning from Errors (DBLE), towards better-calibrated DNNs. DBLE learns a distance-based representation space through classification and exploits distances in the space to yield well-calibrated classification. Our motivation is that a test sample's location in the representation space and its distance to training samples should

---

*Work done during internship with Google Cloud AI.

contain important information about the model's decision-making process, which is useful for guiding confidence estimation. However, in vanilla training, since both training and inference are not based on distances in the representation space, they are not optimized to fulfill this goal. Therefore, in DBLE, we propose to adapt prototypical learning for training and inference, to learn a distance-based representation space through classification. In this space, a test sample's distance to its ground-truth class center can calibrate the model's performance. However, this distance cannot be calculated at inference directly, since the ground truth label is unknown. To this end, we propose to train a separate confidence model in DBLE, jointly with the classification model, to estimate this distance at inference. To train the confidence model, we utilize the mis-classified training samples during the training of the classification model. We demonstrate that on multiple DNN models and datasets, DBLE achieves superior confidence calibration without increasing the computational cost as ensemble methods.

## 2 RELATED WORK

Many studies (Nguyen et al., 2014; Guo et al., 2017) have shown that the classification performance of DNNs are poorly-calibrated under vanilla training. One direction of work to improve calibration is adapting regularization methods for vanilla training. Such regularization methods are often originally proposed for other purposes. For example, Label Smoothing (Szegedy et al., 2016), which is proposed for improving generalization, has empirically shown to improve confidence calibration (Müller et al., 2019). Mixup (Zhang et al., 2017), which is designed mostly for adversarial robustness, also improves confidence calibration (Thulasidasan et al., 2019). The benefits from such approaches are typically limited, as the modified objective functions do not represent the goal of confidence estimation. To directly minimize a confidence scoring objective, Temperature Scaling (Guo et al., 2017), modifies learning by minimizing a Negative Log-Likelihood (Friedman et al., 2001) objective on a small training subset. It yields better-calibrated confidence by directly minimizing an evaluation metric of the calibration task. However, since it requires training set to be split for the two tasks, classification and calibration, it results in a trade-off – if more training data is used for calibration, then the model's classification performance would drop because less training data are left for classification.

Bayesian DNNs constitute another direction of related work. Bayesian DNNs aim to directly model the posterior distribution for a test sample given the training set (Rohekar et al., 2019). In Bayesian DNNs, since getting the posterior of large amount of parameters is usually intractable, different approximations are applied, such as Markov chain Monte Carlo (MCMC) methods (Neal, 2012) and variational Bayesian methods (Blundell et al., 2015; Graves, 2011). Therefore, the calibration of Bayesian DNNs heavily depends on the degree of approximation and the pre-defined prior in variational methods. Moreover, in practice, Bayesian DNNs often yield prohibitively slow training due to the slow convergence of the posterior predictive estimation. (Lakshminarayanan et al., 2017). Recent work on Bayesian DNNs (Heek & Kalchbrenner, 2019) reports that the training time of their model is an order of magnitude higher than vanilla training. As Bayesian-inspired approaches, Monte Carlo Dropout (Gal & Ghahramani, 2016) and Deep Ensembles (Lakshminarayanan et al., 2017) are two widely-used methods since they only require minor changes to the classic training method of DNNs and are relatively faster to train.

## 3 LEARNING A DISTANCE-BASED SPACE FOR CONFIDENCE CALIBRATION

We first introduce how DBLE applies prototypical learning to train DNNs for classification. Then, we explain that under this new training method, a distance-based representation space can be achieved to benefit confidence calibration.

### 3.1 PROTOTYPICAL LEARNING FOR CLASSIFICATION

DBLE bases the training of the classification model on prototypical learning (Snell et al., 2017). Prototypical learning is originally proposed to learn a distance-based representation space for few-shot learning (Ravi & Larochelle, 2016; Oreshkin et al., 2018; Xing et al., 2019). In prototypical learning, both training and prediction solely depend on the distance of the samples to their corresponding class 'prototypes' (referred as 'class centers' in the paper) in the representation space. It trains the model with the goal of minimizing the intra-class distances, whereas maximizing inter-class distances such that related samples are clustered together. We adapt this training method for classification training in DBLE. Subsequent subsections describe the two main components of prototypical training: episodic training and prototypical loss for classification. Later, we explain how to run inference at test time and why it leads to a representation space that benefits confidence calibration.

### 3.1.1 EPISODIC TRAINING

Vanilla DNN training for classification is based on variants of mini-batch gradient descent (Bottou, 2010). Episodic training, on the other hand, samples $K$-shot, $N$-way episodes at every update (as opposed to sampling a batch of training data). An episode $e$ is created by first sampling $N$ random classes out of $M$ total classes[1], and then sampling two sets of training samples for these classes: (i) the *support* set $\mathcal{S}_e = \{(\mathbf{s}_j, y_j)\}_{j=1}^{N \times K}$ containing $K$ examples for each of the $N$ classes and (ii) the *query* set $\mathcal{Q}_e = \{(\mathbf{x}_i, y_i)\}_{i=1}^{Q}$ containing different examples from the same $N$ classes.

$$\mathcal{L}(\theta) = \mathbb{E}_{(\mathcal{S}_e, \mathcal{Q}_e)} - \sum_{i=1}^{Q_e} \log p(y_i | \mathbf{x}_i, \mathcal{S}_e; \theta). \tag{1}$$

At episode $e$, the model is updated to map the query $\mathbf{x}_i$ to its correct label $y_i$, with the help of support set $\mathcal{S}_e$. The model is a function parameterized by $\theta$ and the loss is negative log-likelihood of the ground-truth class label of each query sample given the support set according to Eq. 1.

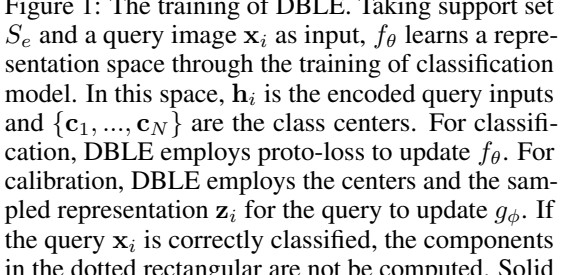

### 3.1.2 PROTOTYPICAL LOSS

How does the support set $\mathcal{S}_e$ help the classification of query samples at episode $e$? We firstly employ the classification model, $f : \mathbb{R}^{n_v} \to \mathbb{R}^{n_c}$ to encode inputs in a representation space, where $\theta$ are trainable parameters. Prototypical training then uses the support set $\mathcal{S}_e$ to compute the center for each class (in the sampled episode). The query samples are classified based on their distance to each class center in the representation space. For every episode $e$, each center $\mathbf{c}_k$ is computed by averaging the representations of the support samples from class $k$:

Figure 1: The training of DBLE. Taking support set $\mathcal{S}_e$ and a query image $\mathbf{x}_i$ as input, $f_\theta$ learns a representation space through the training of classification model. In this space, $\mathbf{h}_i$ is the encoded query inputs and $\{\mathbf{c}_1, ..., \mathbf{c}_N\}$ are the class centers. For classification, DBLE employs proto-loss to update $f_\theta$. For calibration, DBLE employs the centers and the sampled representation $\mathbf{z}_i$ for the query to update $g_\phi$. If the query $\mathbf{x}_i$ is correctly classified, the components in the dotted rectangular are not be computed. Solid arrows represent the forward pass and dotted arrows represent the backward pass to update the network.

$$\mathbf{c}_k = \frac{1}{|S_e^k|} \sum_{(\mathbf{s}_j, y_j) \in \mathcal{S}_e^c} f_\theta(\mathbf{s}_j), \tag{2}$$

where $\mathcal{S}_e^k \subset \mathcal{S}_e$ is the subset of support samples belonging to class $k$. The prototypical loss calculates the predictive label distribution of query $\mathbf{x}_i$ based on its distances to the $N$ centers:

$$p(y_i | \mathbf{x}_i, S_e; \theta) = \frac{\exp(-d(\mathbf{h}_i, \mathbf{c}_{y_i}))}{\sum_{k'} \exp(-d(\mathbf{h}_i, \mathbf{c}_{k'}))}, \tag{3}$$

where $\mathbf{h}_i$ is the representation of $\mathbf{x}_i$ in the space:

$$\mathbf{h}_i = f_\theta(\mathbf{x}_i). \tag{4}$$

The model is trained by minimizing Eq. 1 with $p(y_i | x_i, S_e; \theta)$ in Eq. 3. Through this training, in the representation space that we calculate $\mathbf{h}_i$ and class centers, the inter-class distances are maximized and the intra-class distances are minimized. Therefore, training samples belonging to the same class are clustered together and clusters representing different classes are pushed apart.

### 3.1.3 INFERENCE AND THE TEST SAMPLE'S DISTANCE TO ITS GROUND-TRUTH CLASS CENTER

At inference, we calculate the center of every class $c$ using the training set, by averaging the representations of all corresponding training samples:

$$\mathbf{c}_k^{test} = \frac{1}{|\mathcal{T}_k|} \sum_{(x_i, y_i) \in \mathcal{T}_k} f_\theta(x_i), \tag{5}$$

---

[1]Note that we do not require $N$ to be equal to $M$ because fitting the support samples of all $M$ classes in a batch to processor memory can be challenging when $M$ is very large.

where $\mathcal{T}_k$ is the set of all training samples belonging to class $k$. Then, given a test sample $\mathbf{x}_t$ in the test set $\mathcal{D}_{test} = \{(\mathbf{x}_t, y_t)\}_{t=1}^{N_{test}}$ (where $y_t$ is the ground-truth label), the distances of its representation $\mathbf{h}_t$ (Eq. 4) to all class centers are calculated. The prediction of the label of $\mathbf{x}_t$ is based on these distances:

$$y'_t = \text{argmin}_k\{d(\mathbf{h}_t, \mathbf{c}_k^{test})\}_{k \in \mathcal{M}}, \tag{6}$$

where $\mathcal{M}$ consists of all classes in the training set. In other words, $x_t$ is assigned to the class with the closest center in the representation space. Consequently, for a test sample $(\mathbf{x}_t, y_t)$, if at inference $\mathbf{h}_t$ is too far away from its ground-truth class center $\mathbf{c}_{y_t}^{test}$, it is likely to be mis-classified. Therefore, in this distance-based representation space, a test sample's distance to its ground-truth class center, is a strong signal indicating how well the model would perform on this sample. In the next section, we empirically show it under DBLE and compare with other methods.

### 3.2 DISTANCE TO THE GROUND-TRUTH CLASS CENTER CALIBRATES CLASSIFICATION PERFORMANCE IN DBLE

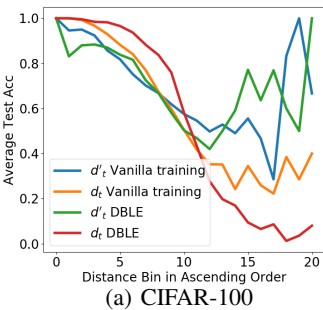
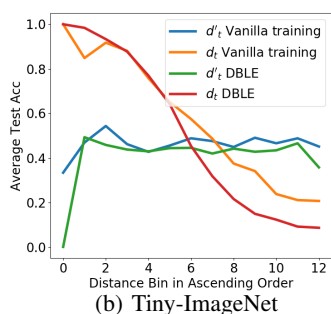

(a) CIFAR-100        (b) Tiny-ImageNet

Figure 2: Average test accuracy as $d_t$ or $d'_t$ increases. $d_t$ is the distance of a test sample $x_t$ to its ground-truth class center and $d'_t$ is its distance to the predicted class center. It shows that $d_t$ in the space achieved by prototypical learning in DBLE can better estimate the model's performance on $x_t$, since DBLE's accuracy curve is more monotonic and less oscillating as the distance increases.

We design an experiment to show that in the space learned by prototypical learning of DBLE, the model's performance given a test sample can be estimated by the test sample's L2-distance to its ground-truth class center. On the other hand, other intuitive measures such as L2-distance to the predicted class center in prototypical learning, L2-distance to the predicted class center in vanilla training or L2-distance to the ground-truth center in vanilla training, can't calibrate performance very well. Here, we describe the empirical observations on DBLE's classification model with prototypical learning, to motivate the confidence modeling of DBLE before we explain the method.

In the learned representation space (either the output space of a model from vanilla training or prototypical learning), for every sample $(\mathbf{x}_t, y_t)$ in the test set $\mathcal{D}_{test} = \{(\mathbf{x}_t, y_t)\}_{t=1}^{N_{test}}$, we calculate its L2-distance to its ground-truth class center $\mathbf{c}_{y_t}^{test}$ as:

$$d_t = d(f_\theta(\mathbf{x}_t), \mathbf{c}_{y_t}^{test}) \tag{7}$$

where $\mathbf{c}_{y_t}^{test}$ is given by Eq. 5. $f_\theta$ is the trained classification model and $d(,)$ is L2-distance. We can also calculate $\mathbf{x}_t$'s distance to its predicted class center $\mathbf{c}_{y'_t}^{test}$ where $y'_t$ is the predicted label, in both DBLE and vanilla training as comparison. We denote $\mathbf{x}_t$'s distance to its predicted class center as $d'_t = d(f_\theta(\mathbf{x}_t), \mathbf{c}_{y'_t}^{test})$. We then sort $\{d_t\}_{t=1}^{N_{test}}$ or $\{d'_t\}_{t=1}^{N_{test}}$ in ascending order and partition them in $I$ equally-spaced bins. For every bin, we calculate the model's classification accuracy of the test samples lying in the bin. Then, we plot the test accuracy curve as the distance increases. The curve shows the the relationship between the distance and the model's performance on $x_t$, illustrating how well the calculated distance can act as confidence score, calibrating the model's prediction.

As shown in Fig. 2, $d_t$ of DBLE, which is the distance of a test sample $\mathbf{x}_t$ to its ground-truth class center, calibrates the model's performance the best. The curve of $d_t$ under DBLE's prototypical learning, despite its slight oscillation near 0 in the end, decreases monotonically to almost zero. It indicates that farther away the sample is from its ground-truth class center, more poorly the model performs on it. These results suggest that $d_t$ can be used as a gold confidence measure, calibrating

the quality of the model's decision of $\mathbf{x}_t$ for a classification model under prototypical learning. This benefit of DBLE's prototypical learning is due to the distance-based training and inference – if a test sample is farther away from its ground-truth class center, it is more likely to be mis-classified as other classes.

Although we have observed this gold calibration in DBLE enabled by prototypical learning, it requires access to the ground-truth labels of test samples, $d_t$. At inference however, we do not have access to the ground-truth label of $x_t$ at inference and $d_t$ cannot be directly calculated. Therefore, we propose to use a separate confidence model to estimate $d_t$, trained jointly with the classification training of DBLE. It regresses $d_t$ by learning from the mis-classified training samples in DBLE's classification training, described next.

## 4   CONFIDENCE MODELING BY LEARNING FROM ERRORS

In the classification model learned by DBLE, a test sample $\mathbf{x}_t$'s L2-distance $d_t$ to its ground-truth class center (Eq. 7), is highly-calibrated with the model's performance on it, as described in Sec. 3. However, $d_t$ cannot be directly computed without ground-truth labels, which is the case at inference. Therefore, we introduce a confidence model parameterized by $\phi$, to learn to estimate $d_t$ jointly with the training of classification in DBLE.

To train $\phi$, a straightforward option would be using the distances for all training samples, $\{(\mathbf{x}_i, d_i)\}_{i=1}^{|\mathcal{T}|}$. Through this way, $\phi$ can be trained to give an estimate of $d_t$ for the test sample $\mathbf{x}_t$ at inference. However, correctly-classified samples constitute the vast majority during training, especially considering that state-of-the-art DNNs yield much lower training errors compared to test errors (Neyshabur et al., 2014). Therefore, if all data is used, training of $g_\phi$ would be dominated by the small distances of the correctly-classified samples, which would make it harder for $g_\phi$ to capture the larger $d_t$ for the minor mis-classified samples. Moreover, given that we choose $g_\phi$ as a simple MLP with limited capacity for fast training, it becomes more challenging to capture larger $d_t$ from the incorrectly-classified samples if all training samples are used (i.e. the confidence model would underfit mis-classified training data). Therefore, we propose to track all training samples that are mis-classified in episode $e$ during the training of the classification model. We save them in $\mathcal{M}_e = \{(\mathbf{x}_s, y_s), \text{where } y'_s \neq y_s\}$ for each episode $e$ to train $g_\phi$. In our ablation studies in Sec. 5.4, we demonstrate the importance of learning from mis-classified training samples vs. learning from all.

Next, we introduce the training procedure of $g_\phi$ with mis-classified samples for estimation of $d_t$ at inference for test sample $t$. We train $g_\phi$ by sampling from the isotropic Gaussian distribution $\mathcal{N}(\mathbf{h}_s, \text{diag}(\sigma_s \odot \sigma_s))$, where the standard deviation $\sigma_s$ is $g_\phi$'s output given mis-classified training sample $\mathbf{x}_s$. $g_\phi$ is trained to output a larger $\sigma_s$ for $\mathbf{x}_s$ with a larger $d_s$. $g_\phi$ takes the representation $\mathbf{h}_s$ (calculated with Eq. 4) of a mis-classified training sample $\mathbf{x}_s$ in $\mathcal{M}_e = \{(\mathbf{x}_s, y_s)\}$ as input, and outputs $\sigma_\mathbf{s}$ as,

$$\sigma_s = g_\phi(\mathbf{h}_s). \tag{8}$$

To train $g_\phi$, we firstly sample another representation $\mathbf{z}_s$ for $x_s$ from the isotropic Gaussian distribution parameterized by $\mathbf{h}_s$ and $\sigma_s$, $\mathbf{z}_s \sim \mathcal{N}(\mathbf{h}_s, \text{diag}(\sigma_s \odot \sigma_s))$. Then, we optimize $g_\phi$ based on the prototypical loss with $\mathbf{z}_s$, using $\mathbf{z}_s$'s predictive label distribution:

$$p(y_s|\mathbf{x}_s; \phi) = \frac{\exp(-d(\mathbf{z}_s, \mathbf{c}_{y_s}))}{\sum_{k'} \exp(-d(\mathbf{z}_s, \mathbf{c}_{k'}))}. \tag{9}$$

When updating $\phi$ with the prototypical loss given $\mathbf{z}_s$, it encourages a larger $\sigma_s$ when the $d_s$ of $\mathbf{x}_s$ is larger. This is because when $\mathbf{h}_s$ is fixed for $\mathbf{x}_s$ in the space, maximizing Eq. 9 forces $\mathbf{z}_s$ to be as close to its ground-truth class center as possible. Given $\mathbf{z}_s$ is sampled from $\mathcal{N}(\mathbf{h}_s, \text{diag}(\sigma_s \odot \sigma_s))$, if $\mathbf{h}_s$ is farther away from its ground-truth center (which is the case of mis-classified training samples), then it requires a larger $\sigma_s$ for $\mathbf{z}_s$ to go close to it. Fig. 3 visualizes how $\sigma$ changes before vs. after updating $\phi$. The training of DBLE is described in Fig. 1 and in Algorithm 1 in the Appendix A.1. Note that in order to make sampling of $\mathbf{z}_\mathbf{s}$ differentiable, we use the reparameterization trick (Kingma & Welling, 2013).

At inference, for every test sample $\mathbf{x}_t$, we make prediction $y'_t$ with $\mathbf{h}_t$ (see Eq. 6). While for confidence estimation of the predictions, we take advantage of $\sigma_t$. After getting $\mathbf{h}_t$ and $\sigma_t$, we sample multiple $\mathbf{z}_t^u$ from $\mathcal{N}(\mathbf{h}_t, \text{diag}(\sigma_t \odot \sigma_t))$ and average their predictive label distributions as confidence

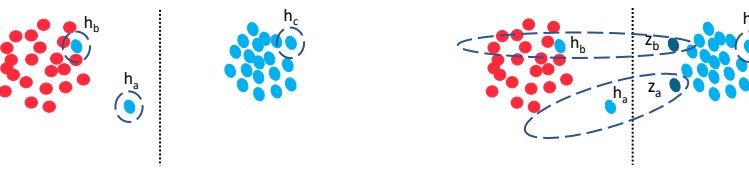

(a) Before updating $\phi$        (b) After updating $\phi$

Figure 3: The Gaussian distribution within one standard deviation $\sigma$ away from mean, shown before and after updating $\phi$. The dotted circles represent the $\sigma$ of the sample inside it. Red and blue dots represent training samples belonging two different classes in the representation space. The dotted line is the decision boundary in the space. Let's take the two mis-classified training samples $\mathbf{h}_a$, $\mathbf{h}_b$ and the correctly classified $\mathbf{h}_c$ as examples. (a), before updating $\phi$, the $\sigma$s of both correctly and wrongly located samples are initialized with a small value. (b), after updating $\phi$, $\sigma$ of mis-classified samples are much larger. Because the proto-loss for calibration will move $z$, sampled from $\mathcal{N}(\mathbf{h}, \text{diag}(\sigma \odot \sigma))$, to be as close to the correct class center as possible.

estimation:

$$\hat{p}(y'_t|\mathbf{x}_t; \phi) = \frac{1}{U} \sum_{u=1}^{U} \frac{\exp(-d(\mathbf{z}_t^u, \mathbf{c}_{y'_t}))}{\sum_{k'} \exp(-d(\mathbf{z}_t^u, \mathbf{c}_{k'}))}. \tag{10}$$

$U$ is the total number of representation samples $\mathbf{z}_t$. $\hat{p}(y'_t|\mathbf{x}_t; \phi)$ is used as the confidence score calibrating the prediction $y'_t$. Through this way, for a test sample farther away from its ground-truth class center (which means they are more likely to be mis-classified), the model will add more randomness to the representation sampling since its estimated variance from $g_\phi$ is large. More randomness in the softmax input space leads to a lower expected softmax output (Gal & Ghahramani, 2016), which is the confidence score $\hat{p}(y'_t|x_t; \phi)$.

## 5 EXPERIMENTS

In this section, we first compare DBLE to recent baselines on confidence calibration of DNNs. We conduct experiments on a variety of data sets and network architectures and evaluate DBLE's performance on two most commonly used metrics for confidence calibration: Expected Calibration Error (ECE) (Naeini et al., 2015) and Negative Log Likelihood (NLL) (Friedman et al., 2001). Results show that DBLE outperforms single-modal baselines on confidence calibration in every scenario tested. We then conduct ablation study to verify the effectiveness of the two main components of DBLE. Implementation, training details, and the details of the baselines are described in Appendix.

### 5.1 EXPERIMENTAL SETUP

**Baselines.** We compare our method with 5 baselines that use a single DNN: vanilla training, MC-Dropout (Gal & Ghahramani, 2016), Temperature Scaling (Guo et al., 2017), Mixup (Thulasidasan et al., 2019), Label Smoothing (Szegedy et al., 2016) and TrustScore (Jiang et al., 2018). We also compare DBLE with Deep Ensemble (Lakshminarayanan et al., 2017) with 4 types of DNNs.

**Datasets and Network Architectures.** We conduct experiments on various combinations of datasets and architectures: MLP on MNIST (LeCun et al., 1998), VGG-11 (Simonyan & Zisserman, 2014) on CIFAR-10 (Krizhevsky et al., 2009), ResNet-50 (He et al., 2016) on CIFAR-100 (Krizhevsky et al., 2009) and ResNet-50 on Tiny-ImageNet (Deng et al., 2009).

**Evaluation Metrics.** We evaluate DBLE on model calibration with Expected Calibration Error (ECE) and Negative Log Likelihood (NLL). ECE approximates the expectation of the difference between accuracy and confidence. It partitions the confidence estimations (the likelihood of the predicted label $p(y'_t|x_t)$) of all test samples into $L$ equally-spaced bins and calculates the average confidence and accuracy of test samples lying in each bin $I_l$:

$$\text{ECE} = \sum_{l=1}^{L} \frac{1}{|\mathcal{D}_{test}|} | \sum_{x_t \in I_l} p(y'_t|x_t) - \sum_{x_t \in I_l} \mathbf{1}(y'_t = y_t)|, \tag{11}$$

where $y'_t$ is the predicted label of $x_t$. NLL averages the negative log-likelihood of all test samples:

$$\text{NLL} = \frac{1}{|\mathcal{D}_{test}|} \sum_{(x_t, y_t) \in \mathcal{D}_{test}} - \log(p(y_t|x_t)). \tag{12}$$

| Method | MNIST-MLP | | | CIFAR10-VGG11 | | |
|---|---|---|---|---|---|---|
| | Accuracy% | ECE% | NLL | Accuracy% | ECE% | NLL |
| Vanilla Training | 98.32 | 1.73 | 0.29 | 90.48 | 6.3 | 0.43 |
| MC-Dropout | 98.32 | 1.71 | 0.34 | 90.48 | 3.9 | 0.47 |
| Temperature Scaling | 95.14 | 1.32 | 0.17 | 89.83 | 3.1 | 0.33 |
| Label Smoothing | 98.77 | 1.68 | 0.30 | 90.71 | 2.7 | 0.38 |
| Mixup | **98.83** | 1.74 | 0.24 | 90.59 | 3.3 | 0.37 |
| TrustScore | 98.32 | 2.14 | 0.26 | 90.48 | 5.3 | 0.40 |
| DBLE | 98.69 | **0.97** | **0.12** | **90.92** | **1.5** | **0.29** |
| Deep Ensemble-4 networks | 99.36 | 0.99 | 0.08 | 92.4 | 1.8 | 0.26 |

| Method | CIFAR100-ResNet50 | | | Tiny-ImageNet-ResNet50 | | |
|---|---|---|---|---|---|---|
| | Accuracy% | ECE% | NLL | Accuracy% | ECE% | NLL |
| Vanilla Training | 71.57 | 19.1 | 1.58 | 46.71 | 25.2 | 2.95 |
| MC-Dropout | 71.57 | 9.7 | 1.48 | 46.72 | 17.4 | 3.17 |
| Temperature Scaling | 69.84 | 2.5 | 1.23 | 45.03 | 4.8 | 2.59 |
| Label Smoothing | **71.92** | 3.3 | 1.39 | **47.19** | 5.6 | 2.93 |
| Mixup | 71.85 | 2.9 | 1.44 | 46.89 | 6.8 | 2.66 |
| TrustScore | 71.57 | 10.9 | 1.43 | 46.71 | 19.2 | 2.75 |
| DBLE | 71.03 | **1.1** | **1.09** | 46.45 | **3.6** | **2.38** |
| Deep Ensemble-4 networks | 73.58 | 1.3 | 0.82 | 51.28 | 2.4 | 1.81 |

Table 1: Test Accuracy, ECE and NLL of DBLE and baselines under 4 scenarios.

## 5.2 RESULTS

Table 1 shows the results. In every scenario tested, DBLE is comparable with vanilla training in test accuracy – applying prototypical learning for classification problems does not hurt generalization of models on classification. On confidence calibration, DBLE performs all single-modal baselines on every scenario tested on both ECE and NLL. Moreover, our method reaches comparable results with Deep Ensemble of 4 networks with a smaller time complexity and parameter size. For MNIST-MLP, CIFAR10-VGG11 and CIFAR100-ResNet50, our method outperforms Deep ensemble (4 networks) on ECE. Among other baselines, Temperature Scaling performs the best in every scenario, which is because Temperature Scaling directly optimizes NLL on the small sub-training set. MC-dropout gives the least improvement on confidence calibration, especially on NLL. The potential reason for MC-dropout's limited improvement on NLL can be that when applying dropout at inference, it adds similar level of randomness on mis-classified samples and correctly classified samples. Therefore, the predictive likelihood of the correct labels becomes smaller in general, which leads to worse NLL. Compared to Temperature Scaling, DBLE decouples classification and calibration, therefore achieves better calibration without sacrificing classification performance. Compared to MC-dropout, our model is trained to add randomness on predictive distributions for mis-classified samples specifically, thus achieving significantly better calibration.

## 5.3 COMPUTATIONAL COST

DBLE adds extra training time complexity and trainable parameters compared with vanilla training. However, compared with Deep Ensembles and other Bayesian methods, DBLE's cost is significantly less. For training time complexity, although DBLE requires two forwards passes for $f_\theta$ at each update step (see Algorithm 1), its actual training time is less than twice of vanilla training since the number of iterations required until convergence is smaller. Empirically we observe that DBLE's total training time on CIFAR10-VGG11 is 1.4x of vanilla training and on CIFAR100-ResNet50 is 1.7x of vanilla training. While Deep Ensemble of 4 networks takes 4x vanilla training time. DBLE adds extra trainable parameters $\phi$ compared with vanilla training. $\phi$ is the parameters of a MLP in practice. It's size is usually $1\% - 3\%$ of the classification model, while ensembling 4 DNNs in Deep Ensemble increases the size of trainable parameters to 4x of vanilla training.

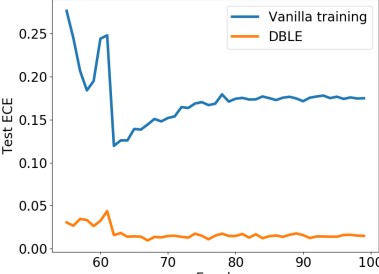 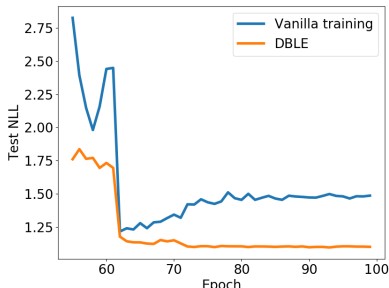

Figure 4: Test ECE and NLL of the last 45 epochs during training on CIFAR-100. The learning rate of both vanilla training and DBLE is annealed by 10x at epoch 60, 70, 80.

| Method | CIFAR100 | | Tiny-ImageNet | |
|---|---|---|---|---|
| | ECE% | NLL | ECE% | NLL |
| Vanilla Training | 19.1 | 1.58 | 25.2 | 2.95 |
| Learning with errors in vanilla training | 18.3 | 1.43 | 20.9 | 2.61 |
| DBLE with calibration learning using all samples | 18.9 | 1.54 | 24.8 | 2.87 |
| DBLE | **1.1** | **1.09** | **3.6** | **2.38** |

Table 2: Ablation study on CIFAR-100 and Tiny-ImageNet

## 5.4 ALGORITHM ANALYSIS AND ABLATION STUDY

**DBLE alleviates the NLL overfitting problem.** NLL overfitting problem of vanilla training has been observed in (Guo et al., 2017) – after annealing the learning rate, as the test accuracy goes up, the model overfits to NLL score (test NLL starts to increase instead of decreasing or maintaining flat). Fig. 4 shows that this phenomenon is alleviated by DBLE. Under vanilla training, the model starts overfitting to NLL after the first learning rate annealing. DBLE on the other hand, decreases and maintains its test NLL every time after learning rate annealing. We see the same trend for test ECE as well – with vanilla training the model overfits to ECE after learning rate annealing while DBLE decreases and maintains its test ECE.

**Distance-based representation space and learning from errors are both essential for DBLE.** We conduct ablation studies to verify the effectiveness of the two main design choices of our model, the distance-based space and learning calibration from training errors. Table 2 shows the results. In "Learning with errors in Vanilla training", we conduct calibration learning with errors in vanilla training. In other words, instead of updating $\phi$ by maximizing Eq. 9, we update $\phi$ by maximizing the softmax likelihood with $\mathbf{z}$ as logits and $f_\theta$ is updated with vanilla training. In "DBLE with calibration learning using all samples", we update $\phi$ with all training samples by maximizing Eq. 9. We can see from the results that firstly, learning to calibrate with errors also helps vanilla training. It improves calibration in vanilla training slightly since it to some extent, introduces more randomness in the decision making process for misclassified samples. However, the improvement is very small without the distance-based space. Secondly, we notice that in DBLE if we learn calibration with all training samples, it significantly decreases DBLE's performance on calibration. The potential reason is the model's underfitting to mis-classified training samples in learning to estimate the distance. Auxiliary analysis on these two can be found in Appendix A.3.

## 6 CONCLUSION

Confidence calibration of DNNs, which has significant practical impacts, still remains an open problem. In this paper, we have proposed Distance-Based Learning from Errors (DBLE) to try to solve this problem. DBLE starts with applying prototypical learning to train the DNN classification model. It results in a distance-based representation space in which the model's performance on a test sample is calibrated by the sample's distance to its ground-truth class center. DBLE then trains a confidence model with mis-classified training samples for confidence estimation. We have empirically shown the effectiveness of DBLE on various classification datasets and DNN architectures.

## 7 ACKNOWLEDGMENTS

Discussions with Kihyuk Sohn are gratefully acknowledged.

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

# A APPENDIX

## A.1 ALGORITHM OF ONE UPDATE OF DBLE

---

Algorithm 1: One update of DBLE. $M$ is the total number of classes in the training set, $N$ is the number of classes in every episode, $K$ is the number of supports for each class, $K_Q$ is the number of queries for each class.

---

**Input**: Training set $\mathcal{D}_{\text{train}} = \{(\mathbf{x}_i, y_i)\}_i, y_i \in \{1, ..., M\}$. $\mathcal{D}_{\text{train}}^c = \{(\mathbf{x}_i, y_i) \in \mathcal{D}_{\text{train}} \mid y_i = c\}$.
\# Build training episode $e$
\#\# Select $N$ classes for episode $e$
$C \leftarrow RandomSample(\{1, ..., M\}, N)$
\#\# Sample supports and queries for every class in $e$
**for** $c$ in $C$ **do**
    $\mathcal{S}_e^c \leftarrow RandomSample(\mathcal{D}_{\text{train}}^c, K)$
    $\mathcal{Q}_e^c \leftarrow RandomSample(\mathcal{D}_{\text{train}}^c \setminus \mathcal{S}_e^c, K_Q)$
**end for**
\# Compute Loss
\#\# Compute center representation for every class $c$ in $e$
**for** $c$ in $C$ **do**
    $\mathbf{c}_c \leftarrow \frac{1}{|\mathcal{S}_e^c|} \sum_{(\mathbf{s}_j, y_j) \in \mathcal{S}_e^c} f_\theta(s_j)$
**end for**
\#\# Compute prototypical loss for classification
$\mathcal{L}(\theta) \leftarrow 0$
**for** $c$ in $C$ **do**
    **for** $(\mathbf{x}_i, y_i)$ in $Q_e^c$ **do**
        $\mathbf{h}_i = f_\theta(\mathbf{x}_i)$
        $\mathcal{L}(\theta) \leftarrow \mathcal{L}(\theta) + \frac{1}{N \cdot K}[d(\mathbf{h}_i, \mathbf{c}_{y_i}) + \log\sum_k \exp(-d(\mathbf{h}_i, \mathbf{c}_k))]$
    **end for**
**end for**
\#\# Compute confidence loss
$\mathcal{L}(\phi) \leftarrow 0$
\#\#\# Make predictions and track mis-classified training samples
$M_e = \{\}$
**for** $c$ in $C$ **do**
    **for** $(\mathbf{x}_i, y_i)$ in $Q_e^c$ **do**
        $y_i' = \text{argmin}_c\{d(\mathbf{h}_i, \mathbf{c}_c)\}_{c=1}^N$   \#$\mathbf{h}_i = f_\theta(\mathbf{x}_i)$
        **if** $y_t' \neq y_t$ **then**
            $M_e \leftarrow AddTo(\mathbf{x}_i, y_i)$
        **end if**
    **end for**
**end for**
\#\#\# Compute confidence loss with mis-classified training samples
**for** $(\mathbf{x}_s, y_s)$ in $M_e$ **do**
    $\sigma_\mathbf{s} = g_\phi(\mathbf{h}_s)$   \#$\mathbf{h}_s = f_\theta(x_s)$
    $\epsilon \sim N(0, 1)$
    $\mathbf{z}_s = \mathbf{h}_s + \epsilon \cdot \sigma_\mathbf{s}$
    $\mathcal{L}(\phi) \leftarrow \mathcal{L}(\phi) + \frac{1}{N \cdot K}[d(\mathbf{z}_s, \mathbf{c}_{y_s}) + \log\sum_k \exp(-d(\mathbf{z}_s, \mathbf{c}_k))]$
**end for**
\# Update $\theta$ with prototypical loss for classification
$\theta \leftarrow \theta - r \cdot \nabla\mathcal{L}(\theta)$
\# Update $\phi$ with confidence loss
$\phi \leftarrow \phi - r \cdot \nabla\mathcal{L}(\phi)$

---

## A.2 IMPLEMENTATION AND TRAINING DETAILS OF DBLE AND BASELINES

For a fair comparison with baseline methods, in every scenario tested, the network architecture is identical for DBLE and all other baselines. As regularization techniques such as BatchNorm, weight

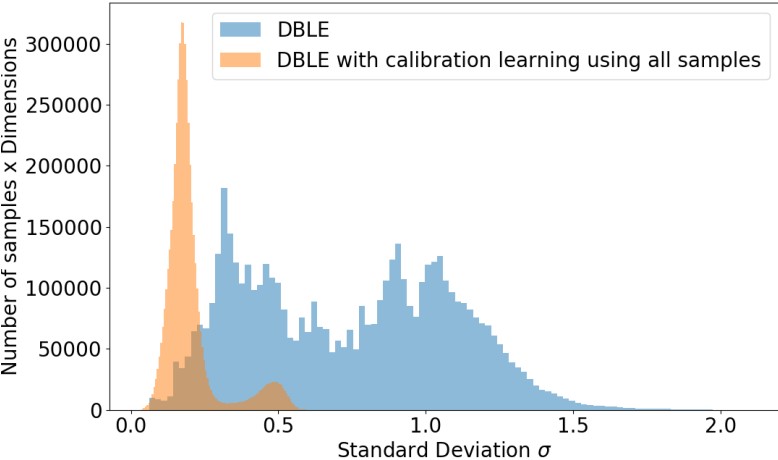

Figure 5: The histogram of $\sigma$ of models trained with all training samples vs. with mis-classified training samples on CIFAR-10.

decay are observed to affect confidence calibration (Guo et al., 2017), we also keep them constant while comparing to baselines in every scenario. All models are trained with stochastic gradient descent with momentum (Sutskever et al., 2013). We use an initial learning rate of 0.1 and a fixed momentum coefficient of 0.9 for all methods tested. The learning rate scheduling is tuned according to classification performance on validate set. All other hyperparameters of classification models for baselines are also chosen based on accuracy on validation set.

There are several unique hyper-parameters in DBLE. We describe how we choose them in the following paragraph. In DBLE, we stop the training when the classification model converges. The confidence model $g_\phi$ is a two-layer MLP with Dropout (Srivastava et al., 2014) added in between. we use ReLU non-linearity (Glorot et al., 2011) for the MLP. We fix the dropout rate as 0.5. The $N$ (number of classes), $K$ (number of shots) and the batch-size of the query set for every episode in DBLE's are firstly tuned according to the classification performance on validation set to reach comparable performance with vanilla training. Then the $N$, $K$ and the batch-size of the query set for every episode are fine-tuned according to the DBLE's calibration performance on validation set. At inference, we fix the number of representation sampling $U$ in Eq. 10 as 20. This is because we empirically observed that a $U$ larger than 20 doesn't give performance improvements. We set the number of bins $L$ in Eq. 11 as 15 following (Guo et al., 2017).

### A.3 Training the confidence with all training samples vs. with mis-classified training samples

In addition to the final results we report in the ablation study (the last two rows of Table 2) comparing training the confidence with all training samples vs. with mis-classified training samples, we also checked statistics of $\sigma$, which is the standard deviation predicted by the confidence model. For CIFAR-10, the mean of $\sigma$ of the model trained with all samples is 0.203, while that of the model trained with mis-classified training samples is 0.740. This suggests that if we train the confidence model with all training samples, the model outputs for test samples are indeed smaller in general, which is aligned with our intuition. The histogram of $\sigma$ under these two methods are shown in Figure 5.

