# OpenReview forum: "Distance-Based Learning from Errors for Confidence Calibration"
_ICLR.cc/2020/Conference — Accept (Poster)_

### Official Review · AnonReviewer1 · 2019-10-22
**Official Blind Review #1**

**Rating:** 6

**Review:**

Summary:
The paper proposes a method to do confidence calibration for deep neural networks. It uses standard
episodic training for prototypical networks, and first shows empirically that the distances of the
embedded test point to its ground truth class center embedding (*not* the predicted class embedding)
are indicative of the confidence of the prediction. Further it proposes to exploit this by training
an auxiliary confidence prediction MLP carefully. To do so they demonstrate that the training needs
to be done of erroneously predicted training examples cf. all the traning examples. They show
results with MLP+MNIST, VGG11+CIFAR10, ResNet50+CIFAR100 and ResNet50+TinyImageNet.


Detailed comments:
The paper is interesting but largely empirical. It shows empirically that:
1. when prototypical networks (and episodic training) is used the distance of test example to true
class center reflects the confidence.
2. this does not hold when `vanilla' training is used
3. an auxiliary MLP can be used to learn to predict this while training only with erroneously
classified training examples cf. all training examples

The results are reported on three networks (MLP, VGG11 and ResNet50) on different benchmarks of
image classification. The confidence prediction improvements wrt baselines are non trivial, while
keeping the accuracy similar, and the computation cost lower than competing methods. Ablations
studies are also convincing.

I would have two broad critical comments on the paper:
1. Would this generalize to other image classification tasks and datasets. Generally distance
(embedding) based networks perform less than softmax based networks on bigger datasets, so an
immediate disadvantage if that happens, is that you would be trading off accuracy cf. vanilla
networks, for better confidence prediction using the required distance based network here.
2. The vanilla training is never formally detailed. I am assuming it was softmax + cross entropy loss
with gradient descent. Would some other loss be helpful? Specially the metric learning based losses like
contrastive or triplet losses come to mind, since they are also distance based. Does the method work
with prototypical networks only or it generalizes to other distance based methods as well?

Minor comments:
The notations are a bit confusing sometimes, and require going back and forth a bit. Eg. \mu is used
for representation of feature (Eq4) while it usually denotes a mean of some sort (so the reader's expectation
could be that it represents class center). Similarly, boldface p is used for class centers, which is
again a bit confusing as being a probability of some sort. In general the notations are different
from the original prototypical networks paper (which I needed to revise); keeping them similar would
help the reader. The contribution of the present paper is more than that anyway.

The erroneously classified training examples are called errors (eg. just before eq8). By errors one
could think that it is a difference between some sort of prediction and the ground truth. Explicitly
calling them erroneously classified training examples would help the reader as well.

The notation \odot is not explained (before eq.9).


**Experience Assessment:**

I have read many papers in this area.

**Review Assessment: Checking Correctness Of Derivations And Theory:**

I assessed the sensibility of the derivations and theory.

**Review Assessment: Checking Correctness Of Experiments:**

I carefully checked the experiments.

**Review Assessment: Thoroughness In Paper Reading:**

I read the paper at least twice and used my best judgement in assessing the paper.

---

> ### Author Response · Authors · 2019-11-14
> **Thanks for the insightful discussions and valuable suggestions!**
>
> Thank you for your thorough reading, insightful comments and valuable suggestions!  We will first address the minor comments and then the two main concerns.
>
> Minor comments:
> 1. Thank you for the suggestions – indeed the \mu and p could cause confusions. We have updated the notation system in the paper.
> 2. We agree.  It also makes sense to avoid using the word ‘training errors’ which people usually use to refer to the measurement of the model’s performance – we will replace.
> 3. We will explain \odot.
>
> Concerns:
>  1. Would the classification performance drop when dataset becomes bigger for prototypical learning?
>
> Datasets with larger number of classes present additional challenges, but it is possible to maintain the generalization performance. We note that the number of classes for CIFAR-100 is 10-fold higher than CIFAR-10, but in both cases DBLE classification performance is comparable to vanilla training.
>
> Among the challenges for higher number of classes, the primary one is the increased compute and hyper-parameter tuning requirements. Hyper-parameter tuning matters when the data set gets bigger. We empirically find that for prototypical learning, larger number of classes within each episode can lead to better classification results. There may be practical limitations on the number of classes that can be sampled at each episode due to the memory limits of the processors, but it can be addressed by straightforward implementation approaches, such as separating each episode into sub-batches to process them in a certain sequential order.
>
> We also note that compared with other distance-based losses such as triplet loss, training classification models with prototypical loss can be more robust to the size of the training set because it optimizes sample-prototype distance instead of sample-pair distances. We explain this in detail under the second question.
>
> Integrating softmax loss together with other distance-based regularizations within our framework could also be an effective future exploration to address this concern. This is because the softmax loss is likely to guarantee the confidence performance without too much effort on hyper-parameter tuning and the distance-based regularization can encourage the effectiveness of $d_t$. We have conducted an experiment to explore this direction which we describe below.
>
> 2. Does the method work with prototypical networks only or does it generalize to other distance based methods as well (e.g. contrastive / triplet losses)?
>
> Yes, by vanilla training we meant softmax + cross entropy loss – we have added clarifications of this in the paper.
>
> Contrastive or triplet losses are distance-based losses, but we think that using them alone won’t solve the problem we discussed under the first main concern. It is reported in multiple papers that models trained with contrastive and triplet losses struggle to reach comparable classification performance with vanilla training and are more complicated to train [1][2]. Moreover, since they optimize pairwise distances of samples, compared to prototypical loss they are also more difficult to optimize.  In contrast, prototypical loss optimizes the distance between query samples and class centers in which the class centers preserve more global information about the class compared to a random sample. This avoids optimizing samples to be overly close to outliers, which simplifies the training.
>
> As a further analysis, we conducted an experiment in which we integrate softmax loss together with triplet loss as regularization. In this experiment, we train the classification model with vanilla training together with triplet loss, and for evaluation we use the distance metric described in Equation 5 and 6. We empirically observe that this gets the same classification performance (without the extra time complexity of episodic training) on CIFAR-10 and CIFAR-100. Moreover, since the classification decisions are still made based on the distances, d_t can still calibrate with the model performance. This proposed method gets NLL score 0.38 on CIFAR-10 and 1.29 on CIFAR-100, which are worse than DBLE (0.09 larger on CIFAR-10 and 0.2 larger on CIFAR-100) but better than vanilla training ( 0.05 smaller on CIFAR-10 and 0.29 smaller on CIFAR-100 ).  This demonstrates that the ‘learning from errors’ and the ‘distance-based evaluation’ components of DBLE can benefit vanilla training plus distance-based regularization methods, but the use of prototypical learning is still a better design choice.
>
>
> We hope that we have fully addressed your questions. Please let us know if you have further comments.
>
> [1]Sohn, Kihyuk. "Improved deep metric learning with multi-class n-pair loss objective." Advances in Neural Information Processing Systems. 2016.
> [2] Schroff, Florian, Dmitry Kalenichenko, and James Philbin. "Facenet: A unified embedding for face recognition and clustering." Proceedings of the IEEE conference on computer vision and pattern recognition. 2015.

---

### Official Review · AnonReviewer3 · 2019-10-23
**Official Blind Review #3**

**Rating:** 6

**Review:**

The paper addresses the (long-standing) problem of classification systems being able to output reliable confidence estimates on its own output. The selected approach is to use the distance to (previously computed) class centers in multi-class classification to help compute the confidence interval. The method is shown to have comparable results on well known datasets but higher efficiency than 2 rival methods: ensemble ANNs and Bayesian neural networks. I would point out that such confidence intervals are all intuitive and have no statistical basis or other independent means of empirical validation. Experienced practitioners are aware of this, but I see that the paper steers wisely clear of overambitious claims. The general intuition of hybrid supervised and unsupervised learning for C.I. (or ellipse) estimation is not new, but an effective and compact representation of this intuition in a DNN context is a valuable contribution, well-placed in the literature - I recommend acceptance.

**Experience Assessment:**

I have read many papers in this area.

**Review Assessment: Checking Correctness Of Derivations And Theory:**

N/A

**Review Assessment: Checking Correctness Of Experiments:**

I assessed the sensibility of the experiments.

**Review Assessment: Thoroughness In Paper Reading:**

I read the paper at least twice and used my best judgement in assessing the paper.

---

> ### Author Response · Authors · 2019-11-14
> **Thanks for the positive comments!**
>
> We appreciate your positive comments that our work is valuable and well-placed contribution in the literature, and your insightful analysis of our paper.  Indeed, instead of making overambitious claims about statistical basis, we focus on improving the predicted confidence to calibrate the model’s performance which can be more objectively evaluated.

---

### Official Review · AnonReviewer2 · 2019-10-23
**Official Blind Review #2**

**Rating:** 6

**Review:**

The paper describes an approach to improve the confidence on deep
neural networks (DNN). The proposed approach uses a distance-based
approach (distance to prototypes) and train using a confidence
model with the classification model using mis-classified examples.

It first learns using a loss function based on pre-computed centroids
of each class evaluated from the training examples. At inference it
assigns the class with the closest center.

To estimate a confidence level, it learns a new model to estimate the
distance using only the misclassified examples.

The authors experimentally showed the benefits of the proposed approach
over several methods.

Finding prototypes and training with distances, on one hand, and
evaluating confidence with Gaussian, on the other, assumes
that the classes are "well defined". Although the authors show
promising results, it is not clear how well the proposed method will
behave in more challenging classification tasks where the classes are
mixed.

Although the authors show that estimating the confidence with
misclassified examples works better I will like to see a further
analysis in the paper.


**Experience Assessment:**

I do not know much about this area.

**Review Assessment: Checking Correctness Of Derivations And Theory:**

I assessed the sensibility of the derivations and theory.

**Review Assessment: Checking Correctness Of Experiments:**

I assessed the sensibility of the experiments.

**Review Assessment: Thoroughness In Paper Reading:**

I read the paper at least twice and used my best judgement in assessing the paper.

---

> ### Author Response · Authors · 2019-11-14
> **Thanks for the comments!  Clarifications and additional results.**
>
> Thank you for the constructive comments!  We address the two main concerns below.
>
> 1. Why does DBLE use misclassified training samples instead of all training samples to train the confidence model?
>
> To answer this question, we first describe our intuitions of training the confidence model with misclassified training samples, and then provide empirical results and statistics that verify the effectiveness of only using mis-classified training samples.
>
> The goal of the confidence model is to capture $d_t$, a sample t’s distance to its ground-truth label in the representation space (illustrated in Equation 7 in the paper), because $d_t$ of a test sample $x_t$ calibrates the model’s performance on $x_t$ in DBLE (see Section 3). A straightforward way to train the confidence model to predict $d_t$ for every test sample would be to use all training samples and their corresponding $d_t$. However, for most deep learning applications, misclassified training samples can be quite rare in the last phase of training. Therefore, if we train the confidence model with all training samples, the correctly-classified samples and their small $d_t$ would dominate the training of the classification model. It may cause underfitting of the mis-classified samples and their large $d_t$. A consequence of this is that the estimation of $d_t$ of test samples would be biased to be small, which leads to overconfidence. Therefore, in DBLE we first initialize the confidence model to make the output $d_t$ for every sample small, and then train the confidence model to fit the large $d_t$ of mis-classified training samples.
>
> We conducted experiments to test this intuition by comparing training the confidence model with only mis-classified samples to training the confidence model with all samples. In addition to the final results we report in the ablation study (the last two rows of Table 2), we also checked statistics of $\sigma$, which is the standard deviation predicted by the confidence model, for the test set under the two training methods. For CIFAR-10, the mean of $\sigma$ of the model trained with all samples is 0.203, while that of the model trained with misclassified training samples is 0.740. This suggests that if we train the confidence model with all training samples, the model outputs for test samples are indeed smaller in general, which is aligned with our intuition.
>
> We agree with the reviewer that a discussion of this question along with the above empirical results would be good to include in the paper.  We have added the discussions above along with the histograms of $\sigma$ in Appendix A.3.
>
>
> 2. How will the proposed method perform when the classes are not well-defined or mixed?
>
> Even if the classes are mixed (e.g., there are two sub-classes under one class label), it is unlikely that there are two clusters under the same class label in the output space for prototypical learning. This is because for every episode the supports of each class are randomly selected, and the representations of the queries of the class are optimized to be close to the single prototype. Therefore the single Gaussian assumption we have for training the confidence model should still be effective even if there is structural information within the class. While we do agree with the reviewer that detecting unsupervised structural information in supervised data is a very interesting and influential research direction for future research, particularly to improve our method for the cases of well-defined or mixed classes.
>
>
> We hope that we have fully addressed your questions. Please let us know if you have further comments.

---

### Public Comment · ~Charles_Corbière1 · 2020-06-26
**Related work of Learning from Errors**

Dear authors,

Thank you for the interesting paper and congratulations on the acceptance at ICLR.

Improving calibration by learning a confidence model from errors is an interesting path. I would like to draw your attention to our work [1] as it shares some similarities with DBLE approach. In our paper, we aim to improve failure prediction, i.e. misclassification detection. Similar to section 3.2 in your paper, we first show empirically that the probability associated to the true class, True Class Probability (TCP), offers a better uncertainty criterion than the standard Maximum Class Probability used in vanilla training. As the true class of a sample is obviously not available at inference, we introduced a confidence model, named ConfidNet, to learn TCP from training data. Such as in DBLE, this confidence model is composed of several fully-connected layers and attached to the penultimate layer of the original classification layer.

As far as I understand, the main differences between your work and ours are the following:
1. The paper address the task of calibration while we focused on misclassification detection ;
2. Confidence model is learned jointly with the classification model in DBLE while ConfidNet is trained on an already trained classification model ;
3. Training is performed only on misclassified samples in DBLE, which seems to be adequate for classification models learned with prototypical learning ;
4. Scalar output of the confidence model in DBLE corresponds to the variance of a Gaussian distribution centered on the sample representation hs. In ConfidNet, we straightforwardly use the output as an uncertainty estimation for misclassification detection.

Please correct me if I have misunderstood anything. Your confidence training scheme combined with prototypical learning is inspiring. I would be very interested to know if this enable to better detect misclassifications.

Best regards,
Charles Corbière

[1] “Addressing Failure Prediction by Learning Model Confidence” Charles Corbière, Nicolas Thome, Avner Bar-Hen, Matthieu Cord, Patrick Pérez. NeurIPS 2019

---

> ### Author Response · Authors · 2020-06-27
> **Thank you for pointing out another related work!**
>
> Hi Charles,
>
> Thank you very much for your interest in our work.
>
> We didn't notice TCP and include it into our related work in the submission because TCP appears available on-line after the submission ddl. NeurIPS 2019's accepted papers are out on NIPS Proceedings website after the ICLR submission ddl. And I checked that the TCP paper is firstly uploaded on arxiv at October 1 2019, which is also after the ICLR submission ddl. I have to make this clear first because I don't want other people to get a wrong impression from our discussion that we deliberately dodged your work when submitting.
>
> I have read TCP carefully and consider it a very interesting empirical observation. In my opinion, the main difference between TCP and DBLE is that the task of confidence calibration has more fine-grained requirements than a binary classification task of misclassification detection. In confidence calibration, the confidence estimation has to ``calibrate’’ the model’s performance on the sample. For example, if you sort all test samples according to their confidence estimations and partition them into bins accordingly, the bin of average confidence estimation 0.5 should have 50% classification accuracy. We empirically observed in the experiment section that under prototypical training and inference, DBLE has this feature (ECE measures that).
>
> I haven’t checked if the true class possibility also has this feature. From Figure 1 in TCP’s paper, it seems that the main chunk of misclassified samples have very low TCP (near 0), which probably means TCP is too binary for confidence calibration. But I think it is worth to check with the confidence calibration exp described in Section 3.2 DBLE paper, how TCP would perform. If TCP doesn’t perform well because most misclassified samples have very low TCP, then an interesting question would be why most misclassified samples have very low TCP under classic training? Is it because the decision boundary is improperly located? Can we add some regularizations during training accordingly to prevent it from happening?
>
> As far as I know, there is no previous work trying to analyze the true class possibility of misclassified samples under classic training. If you could go steps further in this direction and come up with regularizations for classic training to solve confidence calibration problem, the impact could be significantly larger than DBLE. Because DBLE uses prototypical training, which would be slow when the number of classes increases. While a potential regularization method for classic training could be less time-consuming. I would be very happy to see a more time-efficient method defeating DBLE on confidence calibration! :) You can also keep me posted if you want further discussions about new exp results. :)
>
>
> Chen

---

> > ### Public Comment · ~Charles_Corbière1 · 2020-07-01
> > **Thanks for this thorough response**
> >
> > Hi Chen,
> >
> > Thank you for this thorough response.
> >
> > In my opinion, calibration and misclassification detection are two different paths to improve the quality of uncertainty estimates. Equipped with a calibrated confidence measure, a neural network can be incorporated in probabilistic systems where insufficient confidence from one sensor could imply relying more on others. However, calibration doesn't affect the ranking of the confidence estimate on a given test set. On the other hand, improving misclassification detection is beneficial for various applications such as active learning, failure prediction or domain adaptation with self-training.
> >
> > We evaluated calibration of ConfidNet's confidence score in Section 3.2 in the supplementary. While ConfidNet improves ECE score on complex datasets, it remains not as effective as dedicated methods such as temperature scaling. Regarding TCP criterion, we haven't measured its calibration score but its 'binary' behavior' may indeed not be beneficial for confidence calibration.
> >
> > Such as in DBLE, the fact that state-of-the-art DNN yield few training errors may harm ConfidNet training. In supplementary, we tried to 'counter-balance' this effect with Focal loss but it didn't improve misclassification detection. We observed in this case that the confidence network output low confidence estimate for most of the inputs, included correctly-classified samples. On the same matter, a 2-step learning where the classifier weights are trained before the confidence network weights was more stable and yielded better results in our case. Due to the distance-based scores, I guess DBLE approach is more robust to train both networks simultaneously.
> >
> > Again, thank you for taking the time to respond and let's keep in touch for further discussions on these subjects :)
> >
> >
> > Charles

---

### Decision · Program_Chairs · 2019-12-19

**Decision:**

Accept (Poster)

**Comment:**

All reviewers voted to accept this paper.
The AC recommends acceptance.